# Towards a Timepix3 Radiation Monitor for the Accelerator Mixed Radiation Field: Characterisation with Protons and Alphas from 0.6 MeV to 5.6 MeV

Daniel Prelipcean [1,2,*], Giuseppe Lerner [1], Ivan Slipukhin [1,3], David Lucsanyi [1,3], Hampus Sandberg [1], James Storey [1], Pedro Martin-Holgado [4], Amor Romero-Maestre [4], Yolanda Morilla García [4] and Rubén García Alía [1]

1 The European Organization for Nuclear Research (CERN), Esplanade des Particules 1, 1211 Meyrin, Switzerland
2 Physics Department, Technical University of Munich (TUM), Arcisstraße 21, 80333 München, Germany
3 Department of Electrical Engineering and Electronics, University of Montpellier (UoM), Rue Auguste Broussonnet 163, 34090 Montpellier, France
4 Centro Nacional de Aceleradores (CNA—US/CSIC/JA), C. Tomás Alva Edison 7, 41092 Sevilla, Spain
* Correspondence: daniel.prelipcean@cern.ch

**Abstract:** A Timepix3 detector with a 300 μm silicon sensor has been studied as a novel radiation monitor for the mixed radiation field at the Large Hadron Collider at CERN. This work describes a test campaign carried out at Centro Nacional de Aceleradores with quasi-mono energetic protons (alphas) from 0.6 (1) to 5 (5.6) MeV, where orthogonal irradiations are used to obtain an energy calibration, and a low-energy angular scan to estimate the front dead layer thickness of the sensor. The detector is operated in hole collection mode and at a partial bias of 250 μm at 50 V, which increases the charge sharing among pixels to mitigate the signal saturation at high energy depositions. The data, supported by FLUKA Monte Carlo simulations of energy losses in the sensor, show that the Timepix3 monitor operates in a linear regime up to energy depositions of around 600 keV per pixel and 2 MeV per cluster. As a result, the detector has been found to be suitable for measuring charged particle fluxes in the LHC mixed radiation field within the linear calibration regime, with the partial exception of inelastic nuclear reaction hits (mostly from neutrons).

**Keywords:** Timepix3; radiation monitor; Radiation to Electronics (R2E); FLUKA; silicon pixel detector; saturation effect; proton and alpha calibration

## 1. Introduction

The characterization of the radiation field generated by the accelerators at the European Organization for Nuclear Research (CERN) is an essential task to ensure their smooth and reliable operation, preventing radiation-induced failures of critical equipment and electronics. For this purpose, the Radiation to Electronics (R2E) [1,2] effort at CERN employs different types of radiation monitors distributed throughout the accelerator complex, including the beam loss monitor (BLM) system [3], the RadMon system [4,5] and its battery-powered version (BatMon) [6], the distributed optical fibre system (DOFRS) [7], and passive high-level dosimeters [8]. As a possible addition, this work presents a recently developed Timepix3 Radiation Monitor setup, of which the suitability to be used in the scope of R2E radiation level monitoring activities at CERN is being assessed. Timepix detectors have been successfully applied for the following wide range of activities: medical radiotherapy [9], high-resolution photon counting [10], radiation monitoring on the International Space Station [11], 3D colour X-ray [12], radiation imaging [13], luminosity measurements and radiation field characterization in the ATLAS detector [14].

The Timepix technology presents several capabilities that are of interest for R2E monitoring activities at CERN accelerators, among which are the following: (i) the measurement

of single-particle hits with good timing resolution, (ii) the possibility of obtaining information about the direction of incoming particles thanks to the pixel granularity, and (iii) the pixel-by-pixel measurement of the deposited energy with a low minimum threshold, which can be used to obtain particle LET measurements, or combined to obtain the total energy deposition measurements in the full module. By exploiting these features, the foreseen usage at CERN accelerators includes the monitoring of the Total Ionizing Dose (TID) via the total energy deposition measurements, as well as the flux measurement of charged particles. In both cases, the good timing resolution of the Timepix can enable the prompt detection of beam losses, and the directionality information can be used to identify their origin.

This work aims at providing a first characterisation of the Timepix3 Radiation Monitor for CERN accelerator applications. In Section 2, the discussion begins by presenting the key features of the radiation environment in the tunnel of the Large Hadron Collider (LHC) [15], simulated using the FLUKA Monte Carlo code [16–18], CERN distributed, version 4.3.3. The radiation field includes different kinds of particles with broad energy spectra, all originating from the loss of TeV-scale beam particles or secondary collision products [19]. After this, a detailed description of the Timepix3 Radiation Monitor is presented in Section 3, starting from the hardware setup and the Timepix detection principle based on the measurement of the time-over-threshold (ToT) and time-of-arrival (ToA). As described in the section, the charge released by the interaction of ionizing particles within the pixel matrix is typically spread over multiple pixels, leading to the need for a cluster reconstruction algorithm to identify the individual particle hits.

The Timepix calibration procedure that converts the measured ToT per pixel into deposited energy is a key subject of this paper. Several calibration techniques have been studied in the literature, either using internal test pulses [20,21] or, alternatively, using test beams as follows: protons at 8 [22] or 5 [23] MeV, alpha sources below 2 MeV [24–26], low energy gamma ray sources [27], pions at 120 GeV [28], etc. In Section 3, the non-trivial relation between the ToT and the deposited energy per pixel in different energy ranges is presented, involving threshold effects at low energy, a linear regime at intermediate energies, and complex high-energy effects widely discussed in the literature, e.g., in Refs. [22–24]. In principle, high energy effects could be mitigated by operating at a partial bias of the sensor, reducing the amount of collected charge in the core pixel thanks to the enhanced charge diffusion to the adjacent ones [29,30]. For this reason, the Timepix3 Radiation Monitor described in this paper is operated at a partial bias of 50 V, leading to a depletion layer of around 250 μm instead of the full 300 μm thickness of the silicon sensor.

The main results of this paper consist of an experimental campaign dedicated to the calibration of the CERN Timepix3 Radiation Monitor with quasi-mono energetic proton (alpha) beams ranging from 0.6 (1) to 5 (5.6) MeV at the Centro Nacional de Acceleradores (CNA) particle accelerator facility [31,32], following the methods described in Section 4. The measurements included the estimation of the thickness of the dead layer in front of the sensor, which was obtained by performing an irradiation scan of the setup with 597 keV protons at an angle ranging from 0° up to 45°. The results of the test campaign are described in Section 5, starting with a discussion of the cluster reconstruction performance, which included a filtering procedure based on morphological parameters such as radius and density. Subsequently, the cluster-level calibration is presented, as well as the distributions of ToT and cluster size for both proton and alpha beams at different energies. The results of the angular scan for the dead layer estimation are also discussed, along with an investigation of the origin of high-energy saturation effects.

Having established the key features of the Timepix3 Radiation Monitor and its calibration parameters, Section 6 investigates the suitability of the setup for the measurement of the particles from the LHC radiation field, using the FLUKA code to evaluate their expected energy depositions in the sensor. Such energy depositions are discussed in light of the results of the calibration campaign, with special attention paid to the high energy effects described above. Finally, Section 7 is devoted to a summary of the results and

general considerations about the usage of the Timepix3 Radiation Monitor at the CERN accelerator complex.

## 2. The LHC Mixed Radiation Field

The operation of particle accelerators at CERN generates a mixed radiation field that contains different types of particles with broad energy ranges [19]. The composition of the radiation field varies significantly depending on the accelerator under exam, on the position within the accelerator, and on the amount of shielding between the radiation source and the location of interest. When considering the LHC [33], the source of the radiation showers can be the inelastic collisions in the interaction points (IPs), the interactions of the beam with residual gas molecules in the beam pipes, or the interaction of the beam with LHC beam intercepting devices (typically collimators or absorbers).

Figure 1 illustrates an example of the radiation field composition in the LHC tunnel in the proximity of the final focusing magnets at 50 m from IP1, hosting the ATLAS experiment [34], in the form of an energy spectrum in lethargy units simulated with the FLUKA code. In this position, the radiation field originates from the secondary products of inelastic proton–proton collisions in the IP, and the spectrum is obtained in a $20 \times 20 \times 20\,\text{cm}^3$ volume at floor level around 1 m below the accelerator beam line. The radiation field contains charged hadrons (mostly pions and protons) with energies up to several GeV, neutrons with energy extending from the meV range (thermal) to the GeV scale, electromagnetic particles (photons, electrons, positrons) from below the MeV scale to the GeV scale, and muons up to tens of GeV. Instead, the environment is characterised by the absence of heavy ions and by a scarce presence of alpha particles, while the relative contribution of the different particle types and the respective energy distributions can vary depending on the position, the spectrum given in Figure 1 is used as the reference radiation field for which the Timepix3 Radiation Monitor should be optimised.

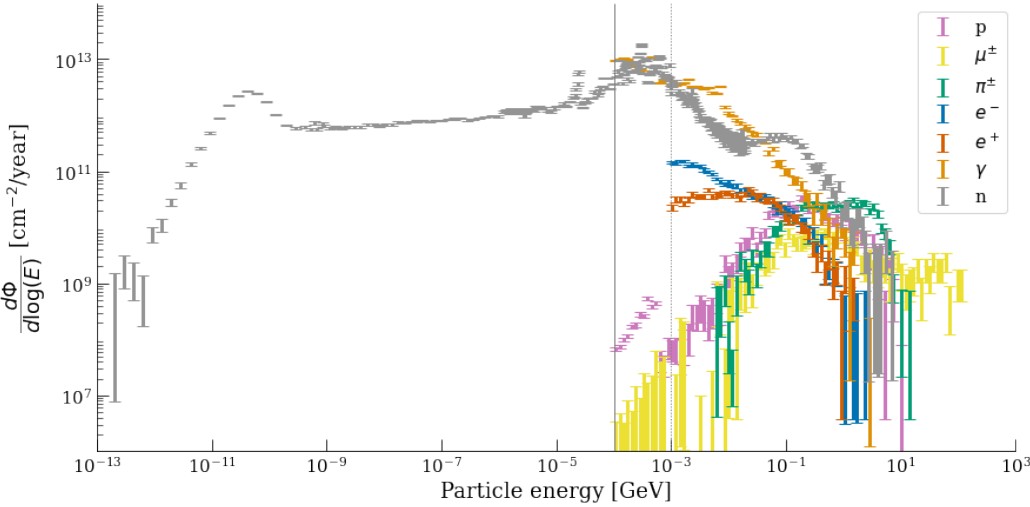

**Figure 1.** Energy spectra (in lethargy units) simulated with FLUKA at floor level below the beam in the LHC accelerator tunnel near the IP hosting the ATLAS experiment for the 2018 operational conditions, right downstream of the inner triplet magnets. The vertical continuous (dotted) grey lines indicate the lower energy threshold set for hadrons and gammas (electrons and positrons) in the simulation, i.e., $10^{-4}$ GeV ($10^{-3}$ GeV) [19].

The existing radiation monitors used for machine protection (BLMs) and for R2E purposes (BLMs, RadMons, DOFRS, and high-level dosimeters) are all capable of measuring the total ionising dose (TID) via different technical solutions [3–8]. The RadMon system can also measure single event effects (SEEs) induced by high energy hadrons (HEH) and thermal neutrons in SRAM memories, calibrated to measure the respective flux, and it also provides measurements of the silicon 1-MeV neutron equivalent fluence via p-in-n

diodes. The Timepix3 Radiation Monitor presented in this work is also designed to be used to measure the TID by summing the energy deposited in its pixels, but it can also be used for charged particle flux measurements. Its excellent time- (1.25 ns) and spatial-resolution (provided by the pixel array with a pixel pitch of 55 µm) make it an optimal instrument to promptly detect radiation showers caused by localised beam losses and, in some circumstances, to provide information about their origin.

## 3. The Timepix3 Radiation Monitor

The Timepix3 Radiation Monitor detector for R2E radiation level measurements at CERN accelerators is based on a setup developed for accelerator beam profile and emittance monitoring [35]. It employs a Timepix3 chip [36] manufactured by ADVACAM [37] in Prague, Czech Republic, part of the Timepix detector family [38], designed by the Medipix collaboration [39]. The Timepix is a hybrid semiconductor pixel detector consisting of a sensor chip with a matrix of $256 \times 256$ pixels, $55 \times 55 \, \mu m^2$ each, which is bump-bonded to the Timepix3 readout Application-Specific Integrated Circuit (ASIC) chip. The detection layer for the Medipix detector family can be made of different semiconducting or semi-insulating materials (such as Si, GaAs, CdTe, CZT, etc.) in combination with the same ASIC chip owing to its hybrid structure. The monolithic sensing layer can have various thicknesses, ranging from 100 to 2000 µm. The Timepix3 Radiation Monitor has a 300 µm thick silicon layer (typically the minimum thickness required to obtain particle identification capabilities), with metalization and dead layers both on the top and on the bottom due to the manufacturing process. The Timepix3 nominal clock time (frequency) is 25 ns (40 MHz), matching the LHC bunch spacing, which could allow a bunch-by-bunch characterization of the radiation environment. This good timing resolution allows the detector to separate individual particle hits that have moderately high flux rates, here, up to approximately $10^8$ particles/(cm$^2$·s).

### 3.1. Setup Description

The Timepix3 Radiation Monitor setup has been conceptually designed into the following three parts: (i) the detector module hosting the Timepix3 chip and sensor, operating in the radiation field to be measured, (ii) the front-end crate, designed as radiation tolerant in order to operate in the proximity (about 5 m away) of the detector module, and (iii) back-end crate, the data acquisition computer and the power supplies, to be placed far enough from the measurement area to be considered safe for electronics and/or human operation.

The full block diagram of the setup is presented in Figure 2. The Timepix3 detector module hosts the Timepix3 hybrid pixel detector mounted on a dedicated board and the power regulators for its operation ($2 \times$ FEASTMP 1.5 V). The board can be powered by a single 8V (5–12V) power supply, and the expected current consumption at this voltage is about 190 mA. For the data (control) signal transmission, the Timepix3 chip uses 8 (10) Low-Voltage Differential Signaling (LVDS) lines, which are routed on the carrier board directly from the chip to six RJ-45 connectors at the front panel of the detector module. To ensure the proper signal quality, category 6a Ethernet cables (double-shielded) are used for the connections between the detector module and the front-end crate. The module has a dedicated detector power supply input, which is a LEMO 00-type connector, located at the front panel. The required power supply depends on the detector type as follows: for the 300 µm p-in-n type detector, a 50 V supply is used, leading to a partial depletion of the silicon sensor. The expected current consumption depends on the detector illumination and temperature, but it should not exceed several mA at 50 V (with the detector covered, it oscillates around 0.1 µA).

The front-end system consists of a single metal crate, responsible for interfacing with the Timepix3 detector module and for the data transmission to the back-end system through optical fibres. Both front-end and back-end use LC-type SFP transceivers. Single-mode optical fibres are used. The front-end system uses FEASTMP power regulators with the allowable input voltage range between 5 V and 12 V. An 8 V power supply with at least

30 W of output power is used, as the expected current consumption at this voltage is roughly 2 A, but can slightly increase for prolonged usage.

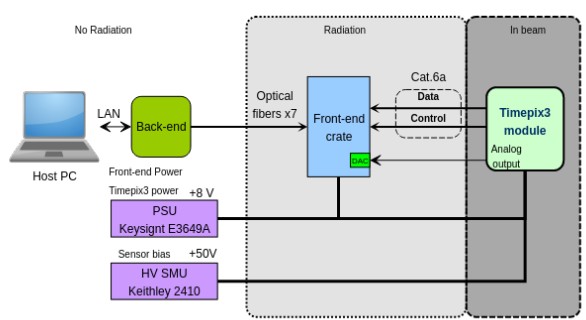 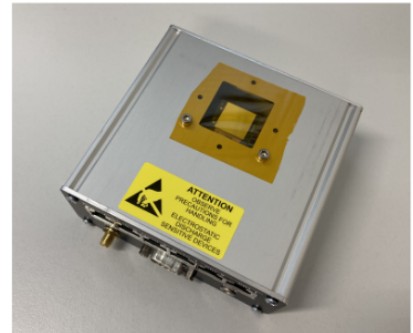

**Figure 2.** (**Left**) Timepix3 setup layout diagram, showcasing the intended operation with the laptop, back-end and power supplies in a no radiation area, the front-end crate that should be radiation tolerant, and the in-beam. (**Right**) Timepix3 detector module, hosting the detector.

The main element of the back-end system is the VC707 development board from Xilinx, hosting the Virtex-7 FPGA. An FMC mezzanine with ten optical fibre transceivers (SFP) is attached to the board and is used for communication with the front-end system. The transceivers' output power is sufficient to operate at distances up to 10 km, but when (as in most cases) the distance is substantially lower, the receivers at the front-end side are saturated since the signal from the front-end does not face enough attenuation. For this reason, attenuators in the range of 3–5 dB are attached to two of the transceivers, where the signal is transmitted in the direction of the front-end crate. A dedicated firmware was written for the back-end FPGA, implementing the necessary procedures for the data retrieval from the front-end system and interfacing with the control PC. The power delivery to the system is realized via a dedicated 12 V AC/DC power converter. The Ethernet interface is used for device communication with the host PC. The readout and processing software on the host PC are based on the PANDA Graphical User Interface (GUI), allowing the setting of the operational parameters as well as data acquisition.

### 3.2. Operational Principle: Signal Formation

When traversed by ionising particles, the silicon sensor of the Timepix3 Radiation Monitor generates output current pulses according to the Shockley–Ramo theorem [40–42] by collecting the free charge carriers (electrons and holes) released in each pixel. In particular, the charge carriers collected by the pixel electrodes are those released in the active portion of the detector with thickness $W$, defined as the region where an electric field $\vec{E}$ is maintained via the application of the bias voltage $V_{bias}$.

The operational principle of the Timepix is that the energy deposited in the pixel is proportional to the number of charge carriers collected by the electrodes, which is, in turn, proportional to the time that it takes to discharge the electrodes via a constant current. The amplified output current pulse of the pixel is shown schematically in Figure 3, along with the timing structure of the Timepix3. When the output signal exceeds a pre-set threshold, a global 40 MHz clock measures the time that it takes for the signal to return below it, i.e., the time-over-threshold (ToT), expressed in clock units (where one time unit corresponds to 25 ns, as determined by the clock frequency). Asynchronously, a local 640 MHz clock starts when the signal exceeds the threshold, thereby defining the fast time-of-arrival (fToA). The clock is stopped by the rising edge of the 40 MHz clock, at which moment the time-of-arrival (ToA) is registered.

The Timepix3 Radiation Monitor uses a p-on-n silicon sensor, with the negative electrodes connect to the analogue front-end, and as such the charge carriers are holes, as shown in Figure 4. Moreover, the module is operated with a partial bias voltage of $V_{bias} = 50$ V, leading to a partially depleted sensor thickness of approximately $W = 250\,\mu m$,

computed with the method described in Ref. [35]. Both these choices lead to a relatively long signal collection time, because the mobility of holes in silicon is three times smaller compared to electrons, and the partial bias voltage leads to a lower electric field $\vec{E}$ in the depleted region. Quantitatively, while the charge collection times can typically be around $\mathcal{O}(10\,\text{ns})$, for the Timepix3 Radiation Monitor it is expected to rise to $\mathcal{O}(100\,\text{ns})$. The longer (longitudinal) charge collection time also allows for more diffusion to occur (at a constant diffusion velocity $v_{dif}$), leading to an increased spread of the charge carriers in the transversal plane. In the case of a pixellated detector structure, this lateral spread causes charge sharing among the chip electrodes, and many pixels could have a signal above threshold; that is, charge carriers generate a cluster of adjacent pixels, as shown exemplarily in Figure 5 for a 5.6 MeV alpha particle hitting perpendicularly the detector, for which a 2D Gaussian shape is typically assumed [29,43,44]. The implications of this feature on the reconstruction of particle hits is further discussed in the next sections.

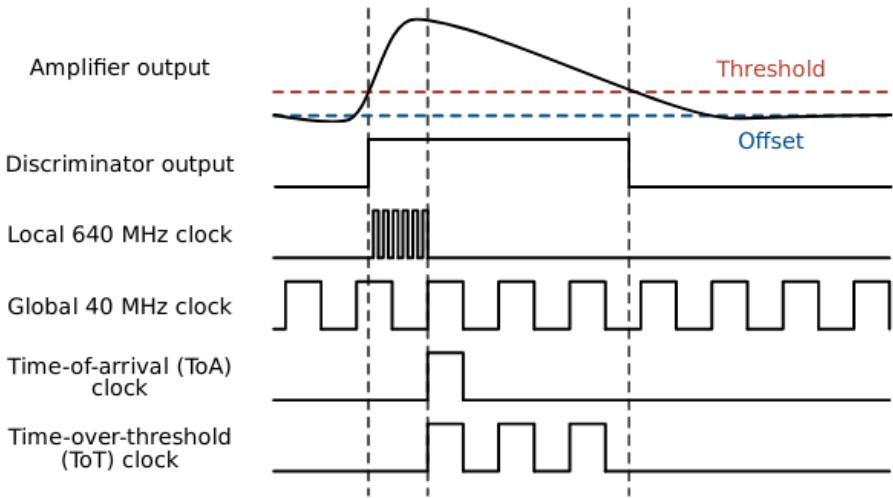

**Figure 3.** Timing diagram for the Timepix3 pixel cell in data driven mode, showing the amplifier output signal, the threshold, and the combined work of the 40 MHz and 640 MHz clocks measuring the time-over-threshold (ToT) and the time-of-arrival (ToA) of the hit.

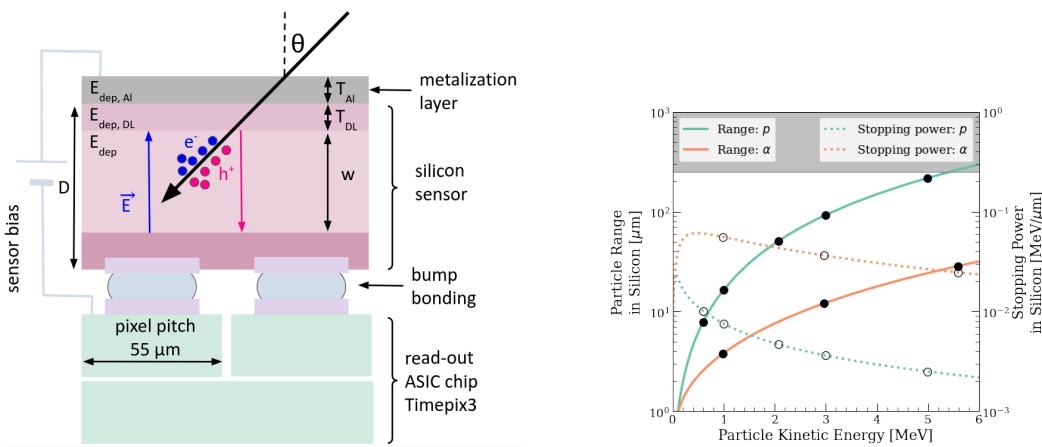

**Figure 4.** (**Left**) Timepix3 detector geometry (not to scale), with a known 500 nm aluminium metalisation layer and an unknown dead layer on top, as well as the depletion layer $W = 250\,\mu\text{m}$ smaller than the detector thickness $D = 300\,\mu\text{m}$, indicating as well the direction of the charge carrier collection. (**Right**) Proton and alpha particles range and stopping power in silicon in the energy range of the 3 MV Tandem accelerator at CNA. Data taken from Ref. [45]. The horizontal grey area starts at 250 $\mu$m, indicating the depletion region thickness.

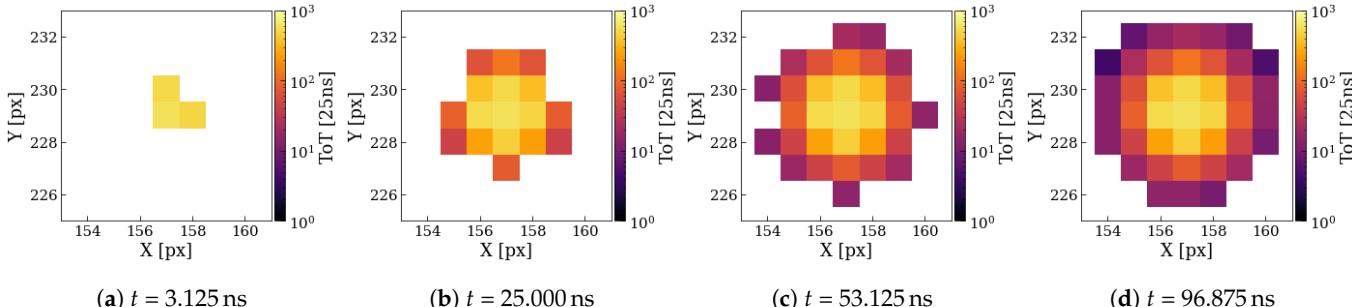

**Figure 5.** Example of a full 5.6 MeV alpha cluster evolution. The particle hits just one pixel at perpendicular incidence in pixel $(x, y) = (157,229)$ and the charges diffuse to adjacent pixels over time, leading to clusters with more pixels. The timing information refers to the combined ToA (40 MHz, or 25 ns) and fast ToA (640 MHz, or 1.5625 ns) counters, when the charge first exceeds the threshold (as shown in Figure 3), and it is uncorrelated with the ToT that is read at the end, when the signal falls below the threshold.

In operation, the Timepix3 Radiation Monitor reads all 65,536 pixels independently of each other. For the applications described in this paper, it is operated in data-driven readout mode, triggering the acquisition of both the ToT and the ToA at every hit, i.e., each time the pixel output signals exceed the threshold.

### 3.3. Clustering

The interaction of a single particle with the Timepix3 Radiation Monitor typically results in a multi-pixel experimental signature, i.e., in more than just one pixel measuring a non-zero ToT. This can be due to (i) particles arriving with a diagonal trajectory with respect to the module, hence crossing more than just one pixel, or (ii) charge carriers released by the incident particle drifting to nearby pixels before being collected by the electrodes [24,43,44]. In both cases, the clusters of pixels from the particle hit must be reconstructed by combining the time and space information via dedicated clustering algorithms [46–48]. For the analysis presented in this paper, a Data Processing Engine (DPE algorithm) [48] developed by Advacam within an ESA project has been used, owing to its computational efficiency for the large data sets collected (millions of particles per configuration).

Figure 5 illustrates the time evolution of a cluster formed by the orthogonal hit of an 5.6 MeV alpha particle on the Timepix3 sensor. At first (i.e., ToA = 0 ns), the charge is released exclusively in the pixel hit by the particle (unless the hit occurs at pixel boundaries, in which case it can be distributed among a maximum of four). Subsequently, due to the charge diffusion process, the nearby pixels begin to collect a portion of the released charge, expanding the dimension of the cluster until the full size is reached (with the external pixel having a ToA that exceeds 100 ns in the case shown). The extent of this effect is determined by the charge collection time, which, in turn, is determined by the applied sensor bias and by the depth of the energy deposition [49,50].

Moreover, if the sensor is partially depleted (as is the case for the Timepix3 Radiation Monitor presented in this work), then the charge collection process is slow and the diffusion effects are enhanced, leading to clusters with more pixels. This feature is exploited in this work to reduce the collected energy per pixel in order to operate in the linear regime of the sensor for the large majority of incoming particles.

When tilted at an angle, the particle tracks inside the detector cross over several pixels. In such cases, the combined effect of the diagonal trajectory of the particle and the charge diffusion process lead to the formation of elongated clusters, which can be exploited to extract information about the direction of the hit.

### 3.4. Pixel Level Energy Calibration Principles

In order to measure the deposited energy in each pixel, a calibration procedure of the Timepix3 Radiation Monitor must be performed, deriving the relation between the output

ToT value in each pixel to the corresponding deposited energy $E_{dep}$ [22,24] (the energy deposited in the sensor can be expressed directly in keV or in the number of electron–hole pairs produced, knowing that the average energy required to produce an electron–hole pair in silicon is ∼3.6 eV), while the exact calibration depends on the operational parameters of the module, notably the threshold level and the bias voltage, general features of the pixel-level ToT vs. energy calibration have been identified in the literature [24], as illustrated qualitatively in Figure 6, where different regions with distinct features can be isolated.

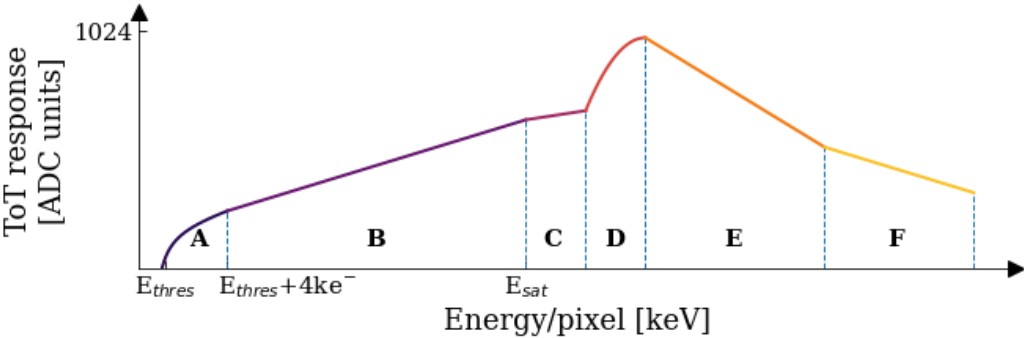

**Figure 6.** Illustration of the pixel-level spectral response of the Timepix, with regions of distinct dependency [24]. The maximum ToT response is limited by the 10-bit registry at 1024 ADC units (here, at 25 ns), while the lower detection limit is operationally set (here, at $E_{thres}$ = 5.3 keV).

At a low deposited energy, the Timepix3 Radiation Monitor measures non-zero ToT only if a threshold $E_{thres}$ is exceeded (set by an operational voltage of 0.57 mV for the detector in this paper, corresponding to a threshold of 5.3 keV). Above $E_{thres}$, there is an initial range (labelled as Region A in the figure) where the ToT response increases in a non-linear way [20,21], typically spanning over a few keV, whereas for higher deposited energies, the Timepix sensors are known to present a wide region of linear response (Region B), which typically extends up to several hundred keV (e.g., 850 keV as found in Ref. [24]). Above a characteristic energy labelled as $E_{sat}$, the ToT calibration curve exhibits a saturation-like effect in the form of a knee (Region C) [22,23,25], above which a parabolic-like increase in the ToT as a function of the deposited energy is reported (Region D) [24]. Lastly, at even higher energies, the ToT is known to decrease as a function of the deposited energy (Regions E and F), leading to ambiguities in the Timepix energy deposition measurements and complex effects [51,52].

Owing to its linearity, Region B represents an ideal range of operation for the Timepix3 Radiation Monitor, whereas all regions below or above it introduce non-linear or multiple-valued effects, particularly when going beyond Region D. In fact, an essential question to be answered by this work is whether the mixed radiation field of CERN accelerators, for which the detector is envisaged, yields energy depositions per pixel that fall (at least for the overwhelming part of the radiation field composition) within the boundaries of Region B, as further discussed in Section 6. For now, it is anticipated that this is mostly the case; therefore, the calibration campaign presented in this paper was specifically targeted at the linear Region B and possibly at the first saturation effects (Region C), neglecting the higher-energy regions as well as the low-energy one (Region A).

### 3.5. Cluster-Level Calibration

Ideally, the calibration curve of the Timepix chip would be obtained by depositing a known amount of energy in a single pixel and measuring the corresponding ToT response. However, as illustrated in Figure 5 for the case of alphas, single particle hits on the sensor lead to the formation of multi-pixel clusters, spreading the deposited energy over several adjacent pixels. This is particularly relevant when the deposited energy is of the order of hundreds of keV or more (i.e., when performing the calibration for relatively high energies), as the number of pixels per clusters increases with the deposited energy.

In the present paper, the calibration of the Timepix3 Radiation Monitor is performed using clusters formed by protons and alphas with known energy that are stopping within the sensors, thereby depositing all their energy within the depleted region, with a fraction lost in the passive layers in front. To perform the pixel-level calibration from multi-pixel clusters, one has to work under the assumption that all pixels are operating within the same linear response (i.e., Region B), by summing the ToTs for all the *N* pixels:

$$ToT_{cl} = \sum_i^N ToT_i = \sum_i^N f(E_i) = \sum_i^N (a \cdot E_{dep,i} + b) = a \cdot \sum_i^N E_{dep,i} + \sum_i^N b = a \cdot E_{dep} + N \cdot b \quad (1)$$

where $ToT_i$ ($E_i$) is the per pixel ToT (energy) value, *a* and *b* are the linear coefficients parametrising the ToT vs. energy curve in Region B, $E_{dep}$ is the total energy deposited (the known beam energy minus the energy lost in the front layers), and *N* is the number of pixels per cluster (which increases with $E_{dep}$). The assumption of having the same linear response for all pixels is tested in Section 5.3, especially for high-energy depositions, for which saturation effects can occur. In fact, the choice of operating the Timepix3 Radiation Monitor with partial bias is intended to avoid concentrating the energy deposition in the centre of the clusters by enhancing the charge sharing, thus ensuring that the conditions of Region B can be met in most (if not all) pixels in the cluster. If these conditions are not met, deviations of the measured calibration curve from Equation (1) can be expected, especially for a high $E_{dep}$ per pixel. Finally, once the calibration curve parameters are established, the Timepix3 Radiation Monitor can be used to obtain energy deposition measurements simply by inverting the expression in Equation (1).

## 4. Materials and Methods

The goal of the work presented in this paper was to characterize the Timepix3 Radiation Monitor, obtaining a pixel-level calibration curve from ToT to deposited energy using high energy hadron beams as described in Section 3, as well as gaining in-house expertise in the operation of the setup to understand its capabilities and limitations. The calibration campaign was carried out by irradiating the detector with quasi-mono energetic ion beams at the Centro Nacional de Aceleradores (CNA) [53,54]. The campaign was performed using 0.6–5 MeV protons and 1–5.6 MeV alphas, a subset of the energy range at this facility. At these energies, all particles stop within the depleted layer of the 300 μm Timepix3 silicon sensor. For the calibration, the effective deposited energy in the active layer of the sensor was derived as the difference between the beam energy and the energy lost in the following two thin layers in front of it: an aluminium metalisation layer of known thickness, and a dead layer of unknown thickness. In addition, to measure the thickness of the front dead layer, an angular scan was performed using 597 keV protons, studying the variation in the effective energy deposited in the sensor. The experimental setup and the calibration procedure, including the dead layer estimation method, are the subject of the next paragraphs.

### 4.1. Experimental Setup at CNA

The test campaign was performed at the Van der Graaf 3 megavolts (MV) Tandem accelerator at CNA, providing quasi-mono energetic protons (alphas) at various energies from 0.6 (1) to 5 (5.6) MeV with an energy spread $\Delta E_{beam} < 3\%$. The full detector module was placed on the specific sample holder into the irradiation vacuum chamber (achieving a pressure in the order of $10^{-6}$ mbar). The Timepix3 Radiation Monitor was masked when the spot beam was focused to set each experimental configuration (i.e., beam energy and flux). Afterwards, the beam was swept using the magnetic scanning system installed in the irradiation beam line, located at 15° from the main line. During the irradiation, a very low particle flux was requested from the facility to minimize the risk of pile-up in the detector module. In this instance, a particle flux lower than $10^4$ particles/(cm²·s) was achieved for these irradiation tests.

The geometry of the Timepix3 Radiation Monitor detector is shown in Figure 4, along with a sketch of the experimental signature of an incoming charged particle at an angle $\theta$ releasing charge carriers before stopping. The module presents a first aluminium metalisation layer on top, with a known thickness of $T_{Al} = 500\,\text{nm}$, followed by a silicon dead layer of an unknown thickness $T_{DL}$. Beyond this, due to the partial depletion of the sensor, the active region does not consist of the entire $D = 300\,\mu\text{m}$ sensor thickness, but only a partial thickness $W = 250\,\mu\text{m}$ (as anticipated in Section 3), while the thickness $W$ depends on the bias voltage and can be calculated, as is performed for the detector under exam, the dead layer is usually unknown due to intellectual property rights of the manufacturing process. A measurement of the dead layer was obtained by performing an angular scan of the deposited energy with a 597 keV proton beam, as further described in the next paragraphs.

Figure 4 also presents the data of proton and alpha range and stopping power in silicon, as extracted from the National Institute of Standards and Technology (NIST) database [45]. Notably, the range of alphas is shorter than the one of protons for the same kinetic energy due to the higher mass. All alpha beams used during the test campaign are hence expected to fully stop within the active layer of the sensor. For protons, assuming an orthogonal incidence on the pixel ($\theta = 0°$), the beam range approaches the thickness $W$ of the active layer when the beam energy is around 5 MeV; beyond this, protons are expected to deposit a significant portion of energy in a non-depleted layer of silicon, or even to pass through the entire pixel thickness without stopping. To avoid this, proton beam energies above 5 MeV were not used during the calibration campaign, as alphas within the available beam energies have significantly shorter ranges.

### 4.2. Energy Calibration and Dead Layer Estimation Strategy

As already discussed, the main scope of the test campaign is to obtain the cluster-level curve of ToT as a function of the deposited energy in the Timepix3 module by performing irradiations with quasi-mono energetic beams directed orthogonally on the sensor (i.e., as shown in Figure 4, with $\theta = 0°$) and stopping within its active layer. For both protons and alphas and for each beam energy, the energy deposited in the active layer of the sensor is effectively given by the difference between the incoming beam energy $E_0$ and the energy lost in the front layers as follows:

$$E_{dep,eff}(E_0) = E_0 - E_{dep,Al}(E_0, T_{Al}) - E_{dep,DL}(E_0 - E_{dep,Al}, T_{DL}) \tag{2}$$

Both front layers are thin, with $T_{Al} = 500\,\text{nm}$ and $T_{DL}$ (to be evaluated as part of the test campaign) expected to be at a similar scale, but the fraction of energy that they absorb can be significant, especially for particles with very short penetration depth like alphas (e.g., at 1 MeV, their penetration depth is just $4\,\mu\text{m}$). The irradiation is performed under vacuum, and thus it is assumed that the beam particle reaches the detector front at its nominal energy $E_0 = E_{beam}$.

The energy deposited in each layer is calculated via Monte Carlo simulations performed with FLUKA, and presented numerically in the results section in Table 1. The source was considered to be a planar wave with an energy distribution of at most 3% (as communicated by the facility), leading to almost delta-peak energy distributions since the particles for all the beam energies stop within the detector. Given the energy scale of the campaign, the following lowest energy thresholds are used: the PRECISIOn FLUKA default, characterized by particles transport thresholds at 100 keV for hadrons (except for neutrons at $10^{-5}$ eV), and lowered threshold for the electromagnetic components as follows: electron/positrons at 1 keV and gammas at 100 eV. Given the small Timepix3 detector geometry, in particular the non-sensitive layers at the sub-micron level, the MULSOPT card was additionally used to enable single Coulomb scattering, thereby deactivating the typical condensed history/multiple Coulomb scattering approach in Monte Carlo codes for computational efficiency.

**Table 1.** Energy depositions $E_{dep}$ simulated using FLUKA in each layer of the Timepix3 Radiation Monitor (500 nm aluminium metalisation layer, 333 nm front dead layer, and 250 µm active volume) for all beam energies $E_{beam}$, together with the total fraction of energy deposited in the front layers $\Delta E = (E_{beam} - E_{dep,eff})/E_{beam}$, as well as the measured average number of pixels per cluster.

| Particle | $E_{beam}$ [MeV] | $E_{dep,Al}$ [keV] | $E_{dep,DL}$ [keV] | $E_{dep,eff}$ [MeV] | $\Delta E$ [%] | No. Pixels |
|----------|------------------|---------------------|---------------------|----------------------|----------------|------------|
| proton   | 0.597            | 32                  | 19                  | 0.543                | 9.2            | 8.2        |
|          | 0.623            | 31                  | 18                  | 0.573                | 6.6            | 8.4        |
|          | 0.996            | 23                  | 14                  | 0.954                | 6.3            | 11.1       |
|          | 2.070            | 13                  | 8                   | 2.047                | 1.0            | 18.2       |
|          | 3.000            | 9                   | 5                   | 2.978                | 0.6            | 21.9       |
|          | 4.997            | 6                   | 3                   | 4.978                | 1.0            | 24.9       |
| alpha    | 0.981            | 169                 | 106                 | 0.706                | 23.3           | 9.6        |
|          | 2.966            | 111                 | 62                  | 2.795                | 10.2           | 22.5       |
|          | 5.595            | 71                  | 43                  | 5.481                | 2.2            | 40.1       |

The measurement of the thickness of the silicon dead layer $T_{DL}$ represents a crucial part of the calibration campaign, and it is obtained by irradiating the sensor with a fixed beam type and energy (597 keV protons) at different angles of incidence (from $\theta = 0°$ to $\theta = 45°$). Under the well-justified assumption that the average energy deposited by the through-going particles in the front layers $E_{dep,front}(E_0, d)$ depends only on the particle energy $E_0$ and on the path length $d$, from simple geometrical considerations the energy lost in the front layers at different angles can be expressed as $E_{dep,front}(E_0, T/\cos\theta)$, where $T$ is the thickness of the layer under exam. Since the particle stops in the depleted layer, one can express the effective energy deposited $E_{dep,eff}(E_0, \theta)$ via the following equation:

$$E_{dep,eff}(E_0, \theta) = E_0 - E_{dep,Al}(E_0, T_{Al}/\cos\theta) - E_{dep,DL}(E_0 - E_{dep,Al}, T_{DL}/\cos\theta) \quad (3)$$

where the dependency of $E_{dep,eff}$ from $\theta$ is solely due to the different fraction of beam energy lost in the front layers, as the particles are then stopping within the active sensor layer regardless of the incident angle.

By assuming a constant stopping power $dE/dx$ both in the aluminium metalisation and the silicon dead layer (i.e., for a small stopping power gradient in these thin layers at these energies). The numerical values and the related level of approximation are presented as follows: the proton stopping power at $E_0 = 0.6$ MeV is $dE/dx|_{Al} = 48$ keV µm$^{-1}$ in aluminium [45], leading to an energy deposition in the $T_{Al} = 0.5$ µm thick aluminium layer varying from $E_{dep,Al}(\theta = 0°) = 31$ keV to $E_{dep,Al}(\theta = 45°) = E_{dep,Al}(\theta = 0°)/\cos 45° = 44$ keV. At this lowered energy of $E' = E_0 - E_{dep,Al}(\theta) = 565$ keV, the stopping power at the end of the aluminium layer increases by 4.8%. Similar values are obtained for silicon, thus Equation (3) becomes the following:

$$E_{dep,eff}(\theta, T_{DL}) = E_0 - \underbrace{\left(\frac{dE}{dx}\bigg|_{Al}(E_0) \cdot T_{Al} + \frac{dE}{dx}\bigg|_{Si}(E') \cdot T_{DL}\right)}_{\text{slope } m} \cdot \frac{1}{\cos\theta} \quad (4)$$

where $E' = E_0 - E_{dep,Al}$. Finally, by irradiating the detector at several angles and plotting the deposited energy $E_{dep,eff}$ vs. $1/\cos\theta$, one can extract the slope $m$, from which the dead later thickness can be obtained as follows:

$$T_{DL} = \frac{m - E_{dep,Al}(\theta = 0°)}{dE/dx|_{Si}(E')} \quad (5)$$

It is clear that the procedure to compute the dead layer thickness $T_{DL}$ via Equation (5) requires a valid calibration of the Timepix3 Radiation Monitor. In turn, the final calibration can only be obtained by a proper evaluation of the energy lost in the dead layer, requiring

the value of $T_{DL}$ as an input. As further described in Section 5, the solution to this puzzle is to set up an iterative process, where a first preliminary calibration of the sensor is derived by assuming $T_{DL} = 0\,\mu m$, and is used to obtain an estimate of $T_{DL}$. The latter value is then used to update the sensor calibration, iterating the process until both quantities converge to a stable result.

## 5. Results

The calibration of the Timepix3 Radiation Monitor is obtained by irradiating the sensor with mono-energetic protons from 0.6 MeV to 5 MeV and alphas from 1 MeV to 5.6 MeV. As shown in Figure 4 (right), for the beam energies explored in this work, both protons and alphas are expected to stop within the active portion of the sensor, and thereby deposit all their kinetic energy into the depletion region (estimated at $W = 250\,\mu m$, for a bias voltage of $V_{bias} = 50\,V$), with the exception of the fraction lost in the front layers. Numerically, the penetration depth of all these beams in silicon varies from 8 μm to 200 μm, while alpha particles have a shorter range (typically one order of magnitude lower compared to protons at the same energy).

The next paragraphs present the full results of the experimental campaign, starting from the event selection and cluster reconstruction (Section 5.1), and continuing with the key quantities measured during the irradiations at different energies (Section 5.2). The calibration of the Timepix3 Radiation Monitor is then iteratively derived in Section 5.3, taking into account the front dead layer, which is also experimentally determined by irradiations at an angle with 597 keV protons. Finally, the observed difference in the saturation levels between protons and alphas is explained.

### 5.1. Event Selection and Cluster Parameters

The first step of the analysis of data from the Timepix3 Radiation Monitor irradiations is the pixel clustering procedure described in Section 3.3, aimed at identifying the results of single-particle hits. Secondly, several parameters are computed for each cluster, in particular, the cluster ToT volume, given by the sum of the individual pixel ToT values, used for calibration purposes. The cluster's shapes are analyzed to define morphological parameters that could allow to distinguish the signal from the background, taking advantage of the information about the experimental test setup (namely, perpendicular irradiation for calibration purposes), as well as the beam particles species (hadrons, leaving round blobs inside the detector, as compared to wiggly lines in the case of electrons).

Before the calibration can be carried out, it is beneficial to define a cluster filtering method to ensure that the clusters from single-particle hits are fully reconstructed and isolated protons, hence removing partially reconstructed clusters and/or pileup events from the analysis. The filtering approach used in this paper is based on the definition of geometric cluster properties [55], which is relevant in this study as follows :

- Cluster size/area $A$: The number of pixels within the cluster;
- Cluster centre of geometry $(c_x, c_y)$: The centre of the cluster, computed as the average point amongst the rows and columns, divided by the cluster area. (Additionally, one could compute the centre of gravity by weighing each pixel by its ToT);
- Cluster radius $R$: The maximum distance between each pixel and the cluster centre of geometry;
- Cluster density $\rho$: Defined as the cluster area divided by the area of a circle ($A = \pi R^2$); therefore, circular clusters will have a density close to unity.

Figure 7 shows the distributions of cluster size, radius, and density parameters, as obtained from the clustered data of 3 MeV protons at CNA. From these quantities, one can define appropriate selections to identify just the nominal clusters, i.e., those corresponding to fully reconstructed single beam particle hits. For instance, by applying a selection on the cluster density (set to $\rho > 0.6$ for all beam energies), one can filter out partially reconstructed clusters or non-hadronic particle species (background), as well as occasional pileup events leading to partially or fully overlapping clusters. By applying an additional

cut on the cluster radius (for 3 MeV protons, $3 < R < 4$), one can further remove the lower energy fragments from the clustering.

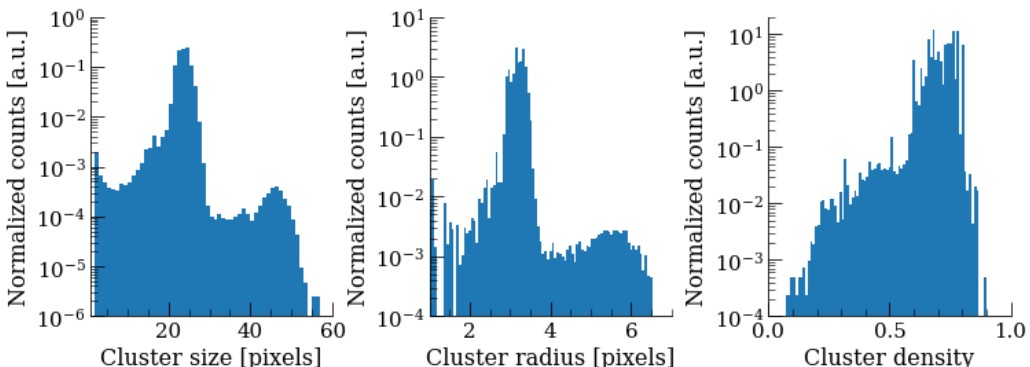

**Figure 7.** Cluster parameters for the 3 MeV protons at perpendicular irradiation.

To assess the performance of the clustering algorithm and the subsequent data filtering procedure, it is helpful to analyse the 2D graphs of the clusters' total ToT volume against the number of pixels (cluster area) before and after the filtering, as shown in Figure 8 for 3 MeV protons. For all beam energies, already before the filtering, a central peak is clearly visible, which can be attributed to clusters from single particle hits. Nevertheless, additional clusters are reconstructed, either with a lower cluster size and ToT volume or with higher values of both quantities. Those clusters with a lower size and/or ToT with respect to the main peak are generally the result of partially reconstructed hits in the clustering algorithm or (less likely) interactions of secondary particles, while the fraction of mis-reconstructed clusters tends to increase for higher beam energies, the filtering procedure has proved to be very efficient in removing them, hence providing a clean data set of well-reconstructed clusters for the calibration analysis. On the other end, dedicated analysis on the clusters with higher ToT volume and size than the main peak has shown that they originate from two (or more) closeby particle hits, resulting in two touching clusters that are reconstructed as a single one by the algorithm, denoted here as clustering pile-up. A physical pile-up in the detector, on the other hand (i.e., a second particle hitting the same pixel before reading the signal of the primary particle), would correspond in such a display to a region of twice the ToT with roughly the same number of pixels per cluster. As evident from Figure 8, there are no such events for the measured flux rates.

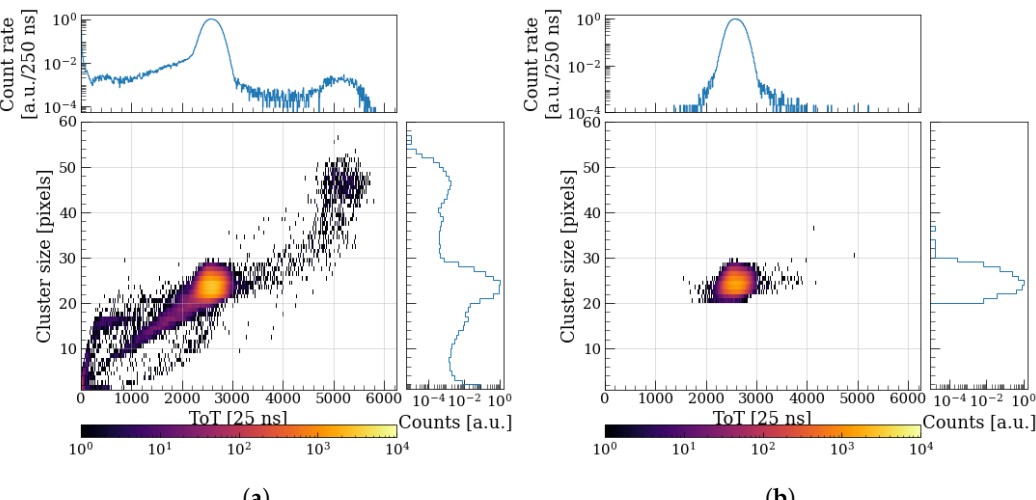

(**a**)                    (**b**)

**Figure 8.** 2D plots of the cluster size against the total ToT per cluster, for protons at 3 MeV (**a**) without and (**b**) with applied filters: $\rho > 0.6$ and $3 < R < 4$.

### 5.2. Proton and Alpha Cluster Measurements

Following the cluster reconstruction and selection process described in the previous section, it is important to examine the key properties of the clusters measured by the Timepix3 Radiation Monitor during the CNA campaign, for all particle types and beam energies with which it was irradiated.

As a first step, Figure 9 illustrates the average shape and the 2D ToT profile of clusters originating from protons and alphas with different energies directed orthogonally on the sensor (i.e., beams for which the angle $\theta$ in Figure 4 is 0°). The clusters have been aligned according to their reconstructed centre of gravity, and the value of the ToT in each pixel represents the average over millions of individual clusters, excluding all pixels with an averaged ToT value below 1 DAC unit (25 ns), such that spurious reconstructed clusters are not unnecesserily emphasized. All clusters are consistently showing larger sizes at higher beam energies, as expected, and the cluster height (i.e., the largest pixel ToT of the cluster) is found in their centre for both protons and alphas at all energies. The latter observation is particularly relevant, because it means that the sensor is not exhibiting the known volcano effect described in the literature for very high energy depositions in Timepix3 detectors [22], where the pixels in the core of the clusters measure a lower ToT as a result of them falling in Regions E and F of the pixel-level calibration curve (see Figure 6). Hence, while at this stage the analysis is only qualitative, the cluster shapes in Figure 9 are already indicating that any saturation effects occurring in the central pixels must not be exceeding (at least on average) Regions C and D of the above curve.

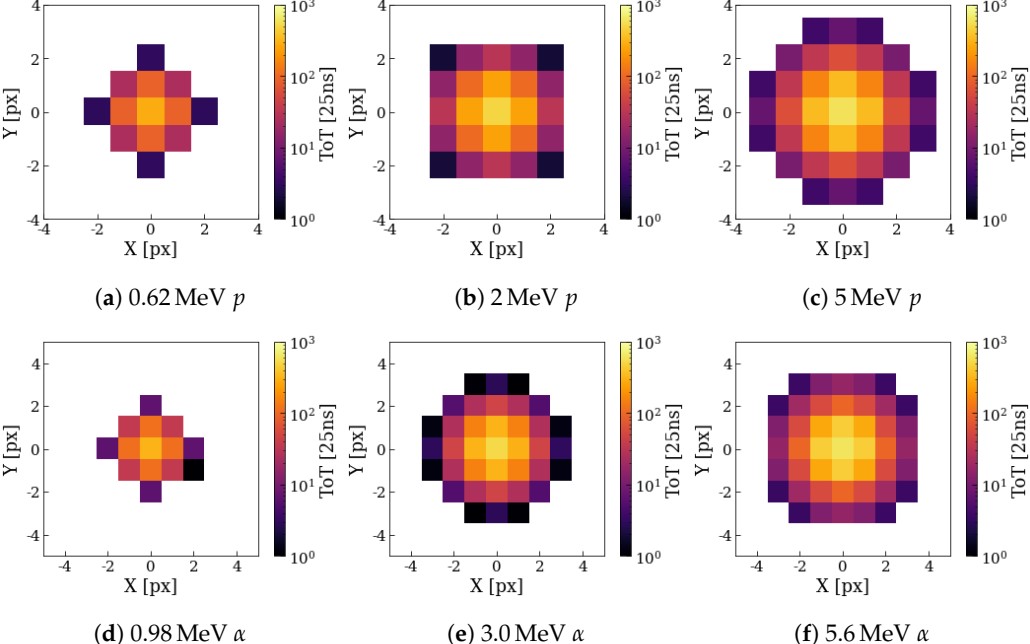

**Figure 9.** Average shapes of the clusters (averaged over millions of reconstructed clusters) for proton (**top row**) and alpha (**bottom row**) runs at different energies, with orthogonal incidence on the Timepix3 Radiation Monitor. Values below 1 ToT unit averaged over the entire data set are set to 0.

After having visualized the cluster shapes, Figure 10 presents the 1D distributions of the total ToT volume of the reconstructed clusters, obtained as the sum of the individual ToT of each pixel in the cluster (as previously shown in Figure 8), for all particle types and energies with which the Timepix3 Radiation Monitor is irradiated orthogonally. All ToT distributions appear as well-resolved peaks, with increasing distribution widths at higher energies, allowing us to fit them with Gaussian functions. The widening of the ToT peaks at high energies can indicate possible resolution effects and non-linearities in the calibration (i.e., saturation), likely in the pixels measuring the highest ToT inside the clusters. In particular, the shape of the ToT peak from 5 MeV protons is showing an extended tail on

the low-ToT side, possibly indicating that a fraction of the protons are reaching the end of the sensitive volume and are thereby not depositing their entire energy in the sensor. Meanwhile, the 5.6 MeV alphas display a widening and flattening of the peak, reducing the quality of the Gaussian fit, and suggesting the presence of saturation effects, as further discussed in the next paragraph. Concerning the cluster size, all curves in Figure 10 show well-defined peaks, with the expected size increase for higher energies.

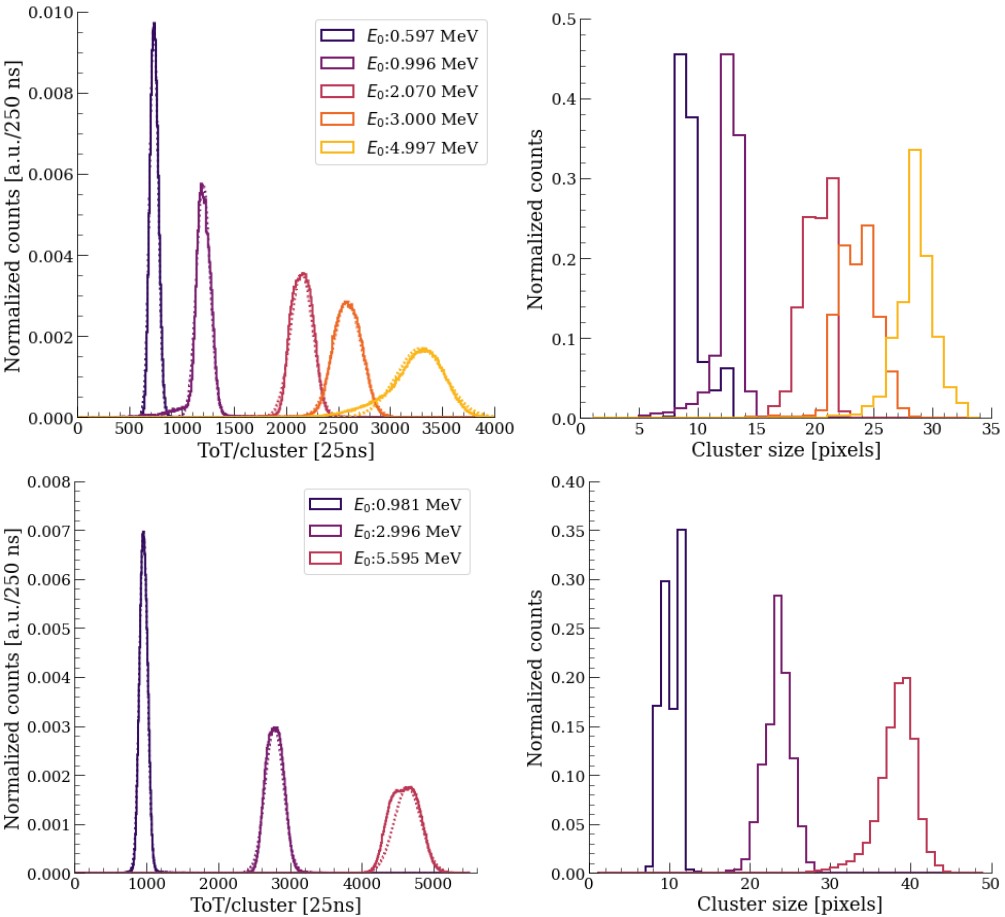

**Figure 10.** Timepix3 Radiation Monitor measurements for (**Top**) proton runs and (**Bottom**) alpha runs with orthogonal incidence and for different energies, displaying the distributions of total cluster ToT (**left**) and number of pixels (**right**) per cluster. The measured peaks (continuous lines) are fitted with Gaussian functions (dashed lines). For the 5.6 MeV alphas, the right slope of the data has been fitted, a choice further motivated in the discussion (Section 6).

Lastly, the distribution of cluster ToT volume of 597 keV protons is plotted again in Figure 11 (left), this time showing all the angles of incidence at which the Timepix3 Radiation Monitor was irradiated, ranging from 0° to 45°. These cluster ToT distributions are similar, but they can nevertheless be distinguished by fitting them with Gaussian functions, which reveal a slight decrease in the mean cluster ToT for larger angles. As anticipated, this angular sweep is necessary for the experimental estimation of the dead layer of the sensor, exploiting the fact that for larger angles, the protons are depositing a larger fraction of energy into it (proportionally to $1/\cos\theta$) before reaching the active volume. The choice of using the lowest available proton energy for the dead layer measurement is motivated by 8 μm proton range at this energy in silicon (as shown in Figure 4), which is comparable in size to the expected dead layer thickness (expected to be around 1 μm or smaller), such that the fraction of energy lost in the dead layer is non-negligible. Indeed, this is what ensures that the mean ToT values of the peaks in Figure 11 are well distinguishable.

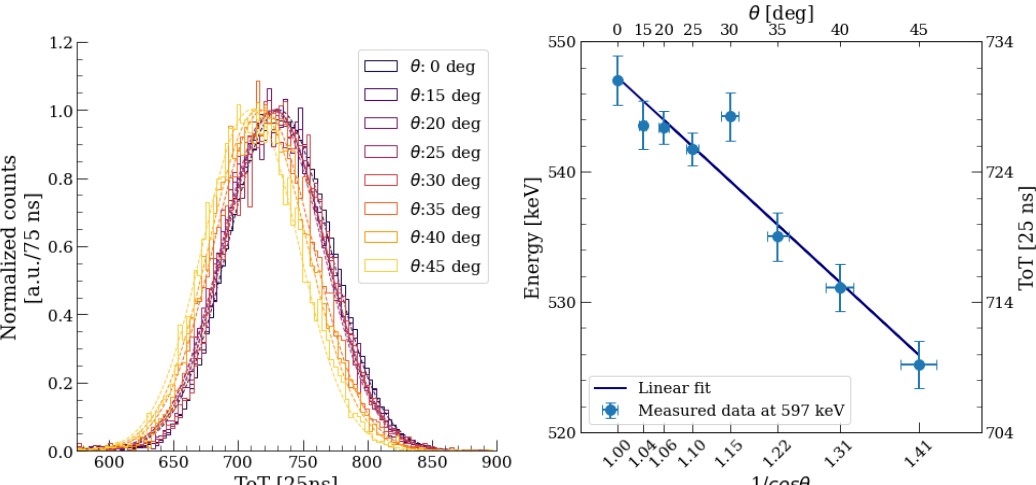

**Figure 11.** (**Left**) Timepix3 measurements for proton runs at 597 keV swiping through incident angles from 0° to 45°, displaying a (**Right**) deposited energy variation that can be fitted with a straight line against the inverse of the irradiation angle cosine.

As outlined in Section 4.2, the dead layer estimation procedure requires the conversion of the mean ToT values of the above peaks into deposited energies using a pre-established calibration. The values are then plotted against the inverse of the cosine of the irradiation angle $\theta$, as shown in Figure 11 (right). By performing a linear fit of this graph, it is then possible to obtain the values of the coefficients of Equation (4) (notably the slope $m$), from which the value of the dead layer can be directly derived. Since this procedure requires a calibration, and in turn, the calibration requires the knowledge of the dead layer thickness, the two quantities are evaluated iteratively, as further outlined below.

### 5.3. Energy Calibration and Dead Layer Estimation

The cluster ToT volume data presented above serve as the basis for the Timepix3 Radiation Monitor calibration, obtained via the procedure outlined in Section 3.5, and supported by the estimation of the silicon front dead layer. Along with the ToT data, FLUKA simulations are employed to quantify the energy deposited $E_{dep}$ in the sensor from a beam of particles $E_0$ via Equation (2), such that one takes into account the energy lost in the aluminium metalisation layer, with a known thickness of 500 nm, and in the dead layer, with a thickness to be evaluated via the angular scan. As previously mentioned, the sensor calibration and the dead layer estimation have been obtained via an iterative process. The calibration is expressed via Equation (1), where the parameters $a$ and $b$ are derived by performing a linear fit that uses only the four lowest energy data points (three for protons and one for alphas, up to 1 MeV) to ensure that the values are not affected by high-energy saturation effects. For all energies, a first calibration is derived by assuming that the thickness of the silicon dead layer is null. This first calibration result is used as an input to obtain a first evaluation of the dead layer, from the data of the angular scan and via Equation (5). Subsequently, the calibration is recomputed using FLUKA simulations with the updated dead layer thickness, repeating the procedure until stable values of $T_{DL}$, $a$, and $b$ are reached, after just six iterations, as shown in Figure 12.

At the end of the iterative process, the linear fit of $E_{dep}$ as a function of $1/\cos\theta$ shown in Figure 11 is the following:

$$E_{dep} \text{ [keV]} = (598 \pm 68) \text{ [keV]} - (51.5 \pm 5.8)\frac{1}{\cos\theta} \text{ [keV]}, \tag{6}$$

where the uncertainty on the fit parameters is included. Using Equation (5), and considering a stopping power of $dE/dx|_{Si}(E_0 - E_{dep,Al}) = 56.45 \text{ MeV μm}^{-1}$ for 0.6 MeV protons in silicon (from [45]), the dead layer thickness is obtained as follows:

$$T_{DL} = 333 \pm 38(\sigma_{fit}) \pm 17(\sigma_{dE/dx}) \pm 4(\sigma_\theta) \pm 1(\sigma_{ToT}) \text{ nm} \tag{7}$$

which includes uncertainties associated with the fit parameters ($\sigma_{fit}$), the assumption of a constant stopping power ($\sigma_{dE/dx}$), for which a 5% error is considered (see Section 4.2), the measurement of the angle ($\sigma_\theta$, assuming a 1% accuracy on $\theta$), and the error on the mean of the Guassian peaks in the measured ToT data ($\sigma_{ToT}$). The measured silicon dead layer of 333 nm is in line with the expectations, as other silicon detectors have dead layers of the order of 500 nm [56,57].

For the above value of the silicon dead layer thickness, the FLUKA code is used to evaluate the deposited energies in each layer of the Timepix3 Radiation Monitor during the irradiations with orthogonal angle, as presented in Table 1 along with the measured average number of pixels per cluster $N$ for each energy and particle type. The simulations indicate that alpha particles are losing a larger fraction of their energy in the front layers of the sensors (up to 23.3%) compared to protons with the same initial energy due to their higher stopping power. The cluster-level calibration is obtained by plotting the mean values of the cluster ToT (obtained by fitting the curves in Figure 10) as a function of the corresponding deposited energies in the active volume of the sensor (i.e., $E_{dep,eff}$ in the table), as illustrated in Figure 13. As saturation effects at high energy are clearly visible in the figure, the linear calibration fit based on Equation (1) is performed using only the four lowest energy data points (i.e., those with beam energy up to 1 MeV) obtaining the following result:

$$ToT_{reg}(E_{dep}) = E_{dep} \cdot \left[ 993 \pm 51(\sigma_{ToT}) \pm 30(\sigma_E) \pm 12(\sigma_{fit}) \right] \left[ \frac{25 \text{ ns}}{\text{MeV}} \right]$$

$$+ N \cdot \left[ 23.10 \pm 1.18(\sigma_{ToT}) \pm 0.69(\sigma_E) \pm 0.27(\sigma_{fit}) \right] \left[ \frac{25 \text{ ns}}{\text{pixel}} \right] \tag{8}$$

where the variable cluster size $N$ is considered. The uncertainties are as follows: $\sigma_{ToT}$ is the averaged $1 - \sigma$ of the measured ToT values per cluster for all the beam energies, $\sigma_E$ is the beam energy uncertainty, evaluated at an average of 3% over all beam energies as communicated by the facility, and $\sigma_{fit}$ is the error on the least squares fit of the calibration curve.

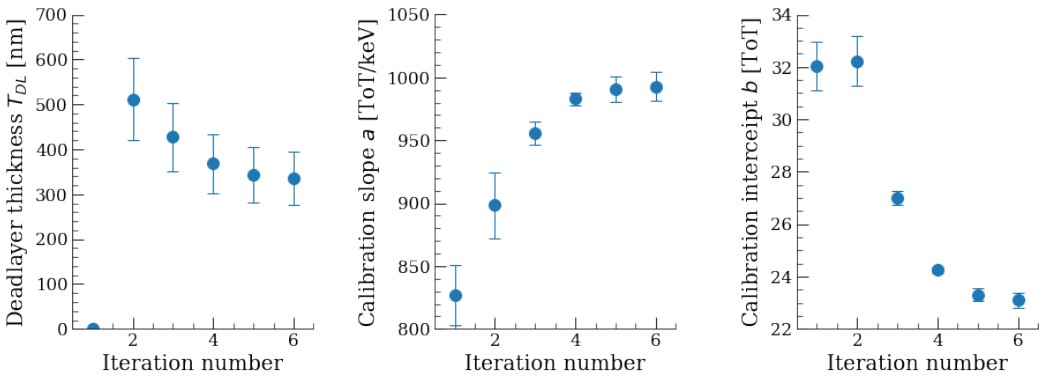

**Figure 12.** Values of the silicon dead layer thickness $T_{DL}$, and of the calibration parameters *a* and *b* from Equation (1), at each step of the iterative data analysis process employed for their evaluation.

The interpolated cluster ToT volume as expected from Equation (8) is included in Figure 13 for all energies, confirming the manifestation of saturation effects at high energies. The saturation is visible for all beam particles above 2 MeV, and more prominent for alphas than for protons, which is explored in the next section. To further explore the origin of the saturation, the distribution of the ToT per pixel is plotted in Figure 14 for all beam energies and particle types. For 2 MeV protons, which represent the lowest-energy beam for which a mild saturation is visible in Figure 13, the pixels with the highest ToT are

measuring between 550 and 600 ToT [25 ns] units. The evidence of saturation effects with this beam indicates that such pixels are presumably already beyond the linear regime of the calibration (i.e., beyond Region B of Figure 6), and likely in the first energy range where saturation is exhibited (Region C). According to the measured calibration (Equation (8)), a ToT range between 550 and 600 ToT [25 ns] units corresponds to a deposited energy of the order of 600 keV per pixel, which is rather consistent with the expectations, as the characteristic pixel-level saturation threshold found in the literature is of 850 keV [24].

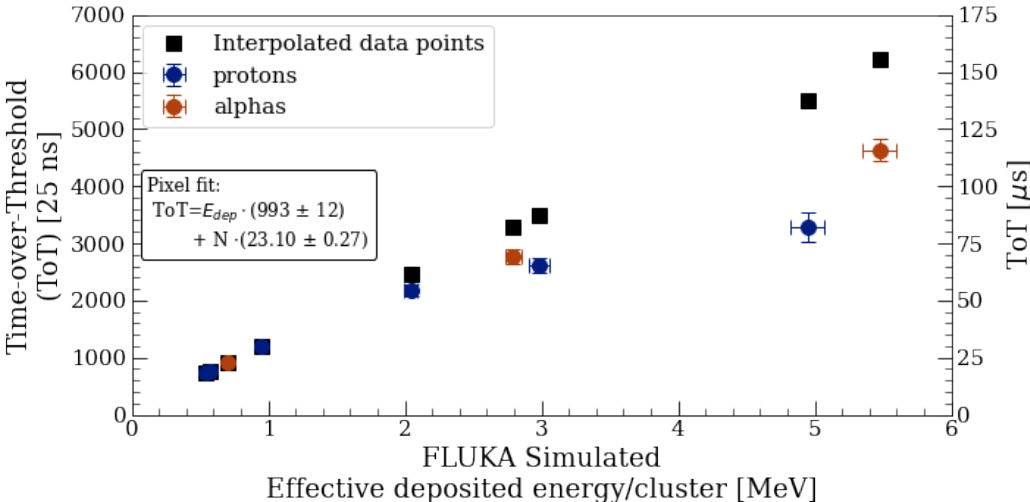

**Figure 13.** Energy calibration per cluster, with the cluster ToT volume plotted against the FLUKA simulated energy deposition in the active sensor, for protons (blue points) and alphas (red points). The interpolated data points (black squares) are computed according to Equation (8), which takes into account the average number of pixels per cluster, taken from Table 1.

Moreover, Figure 14 shows that a few pixels occasionally record high ToT values, beyond the edge of the main distributions, occasionally reaching the maximum digitally measurable ToT (1024 DAC units, given by the 10-bit registry), while these cases are very rare, they could indicate that such pixels have entered the parabolic calibration region (Region D in Figure 6). No evidence of pixels entering Region E and F of the same figure is observed, coherently with the absence of volcano effects in the cluster shapes in Figure 9.

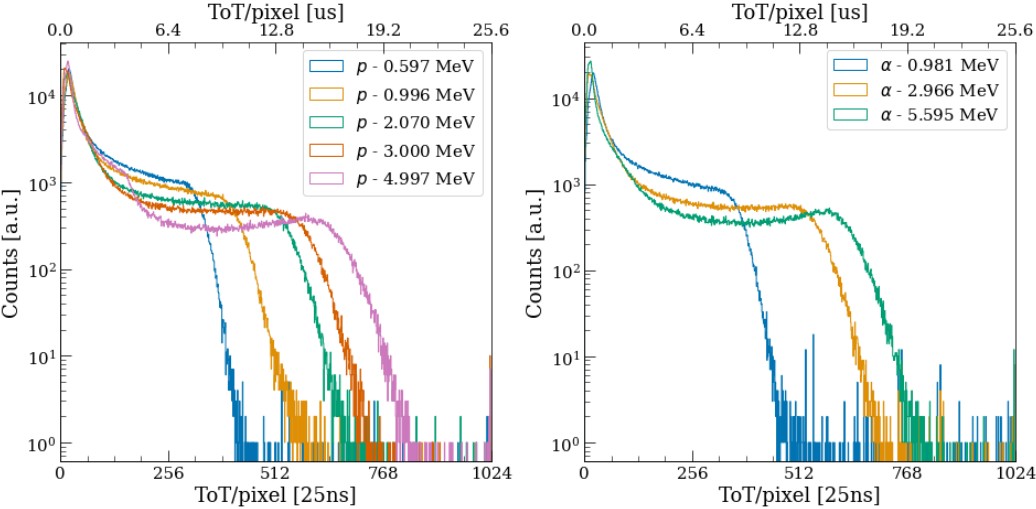

**Figure 14.** Measured distribution of pixel ToT at different beam energies, for (**Left**) protons and (**Right**) alphas. The maximum ToT is given by the 10-bit registry as $2^{10} = 1024$ ToT units.

Having found that saturation effects at pixel level begin between 500 and 600 ToT [25 ns] units, one can further analyze the broadened ToT volume peak observed in Figure 10 for 5.6 MeV alphas. Following a method used in [22], the left graph in Figure 15 presents the breakdown of the distribution of the cluster ToT volume for 5.6 MeV alphas, divided into four categories based on the number of "hot" pixels that are exceeding a ToT threshold set to 565 [25 ns] units. Since one can assume with good approximation that all alphas are depositing an equivalent amount of energy in the sensor, it is reasonable to associate these categories to a variable degree of charge sharing between the central pixels, linked to the position of the alpha particle hit within the struck pixel. In practice, when the hit occurs in the centre of a pixel, a larger fraction of the released charge remains within its boundaries, leading to a single hot pixel in the cluster, whereas if the hit occurs close to a pixel boundary, the charge sharing is enhanced, resulting in up to four hot pixels. As Figure 10 clearly shows that the distribution with only one hot pixel presents a lower mean cluster ToT compared to the others, it is fair to attribute the shift to a higher level of saturation in the central pixel, due to the higher fraction of alpha energy which is deposited into it, while this interpretation involves hypotheses that cannot be easily verified in the data, it provides interesting insights into the origin of the broadening of the cluster ToT peak of 5.6 MeV alphas, linking it directly to saturation effects.

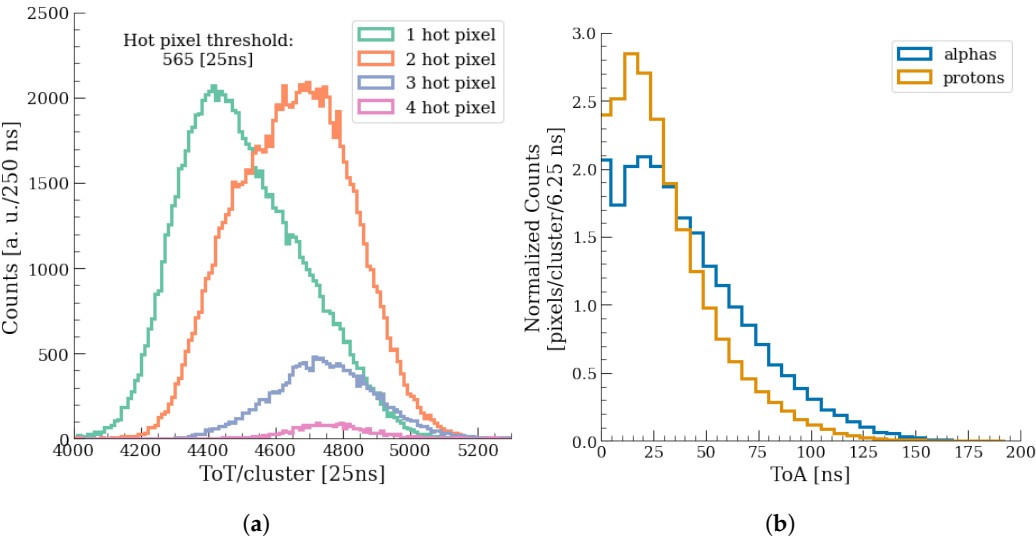

(**a**)                                     (**b**)

**Figure 15.** (**a**) Distribution of cluster ToT volume for 5.6 MeV alphas with variable number of pixels exceeding a threshold set to 565 [25 ns] ToT units, and (**b**) distribution of pixel ToA in 3 MeV proton and alpha clusters, where the bin at ToA = 0 corresponds to the first pixel that is hit, and the bin width is set to four times the fToA timing resolution of 1.5625 ns.

Lastly, it is thorough to investigate why the level of saturation observed for protons in Figure 13 is larger than the one for alphas. As a first step, one can compare the cases of 3 MeV protons and 2.966 MeV alphas in Table 1, noting that despite a larger fraction of energy lost in the front layers (which results in less energy being measured by the sensor), the alphas result in slightly larger clusters (22.5 average pixels per cluster versus 21.9). An explanation of this feature stemming from the charge collection within the silicon detector is that the alpha hits lead to more charge sharing amongst pixels as opposed to protons with similar kinetic energy. This is presumably due to their different linear energy transfer (LET), leading to a different distribution of the energy deposition in the active volume of the sensor (visible in Figure 4), ultimately leading to different ranges in silicon. In turn, a difference in the amount of charge sharing determines the level of saturation at the pixel level, leading to the observed differences between proton and alpha runs.

To further verify this, the right graph in Figure 15 presents the distribution of the time-of-arrival (ToA) of the pixel hits, measured with the fToA timing resolution of 1.5625 ns, for clusters formed by 3 MeV protons and alphas (recalling that the actual energy deposited

in the active volume is lower than the beam energy, especially for alphas, as shown in Table 1). Since most of the charge is deposited closer to the front layer (and further from the collection electrode) for alphas (due to the shorter range at the same energy) than for protons, the charge collection time is also larger. In turn, this prolonged time allows for more charge sharing to occur, which happens at a constant diffusion speed. Within the clusters, it is straightforward to associate larger amounts of charge sharing between pixels with a longer average amount of time taken by the charge to propagate to all pixels. Indeed, the average ToA of pixel hits in alpha clusters is larger compared to the case of proton clusters, fully confirming the above interpretation.

## 6. Discussion

Having thus calibrated and characterised the Timepix3 Radiation Monitor, it is at this stage relevant to assess its suitability to perform measurements in the CERN accelerator complex, e.g., in the LHC radiation field described in Section 2. Using the FLUKA model of the detector, additional simulations are employed to evaluate the energy deposited in the active volume of the sensor by particles with the characteristic energy distributions of the LHC mixed field, i.e., for the spectra shown in Figure 1. To a first approximation, the particles are assumed to be perpendicularly incident on the detector, even though in realistic cases, they may hit it at various angles depending on the radiation source location.

The expected distribution of the deposited energies within the Timepix3 Radiation Monitor is shown in Figure 16 for the most prevalent particle species in the LHC mixed field. The detection efficiency for the charged particles in the silicon sensor is effectively 100%, and the majority of those that would reach the detector in the LHC tunnel have high energies and longer ranges, such that they pass through the entire active volume depositing similar amounts of energy (corresponding to the main peaks in Figure 16), implying that the detector can be used to measure charged particle fluxes, rather than being capable of restoring the full energy spectrum of Figure 16. Charged particles with shorter range can deposit more energy, up to a maximum given by the particle energy for which the range is equal to the active volume thickness (which is evaluated at $W = 250\,\mu m$, for a bias voltage of $V_{bias} = 50\,V$), which is about 5 MeV for protons and about 2 MeV for muons and pions. At these energies, the distributions in the figure are exhibiting visible "knees". Concerning neutral particles, the energy deposition from photons does not exhibit peaks, and it is limited to a maximum of 1 MeV. Neutrons have a low detection efficiency, and when they interact elastically, their energy deposition is also expected to be low.

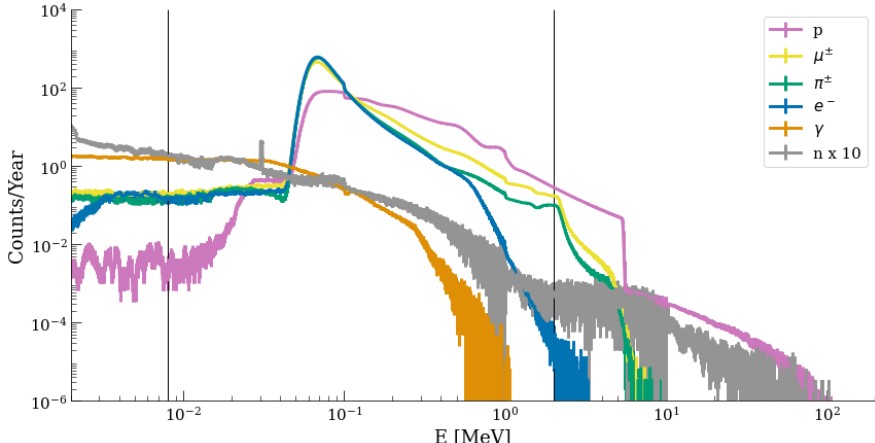

**Figure 16.** Distribution of the energy deposited in the active volume of the Timepix3 Radiation Monitor by the particles composing the LHC mixed radiation field (from Figure 1), simulated with FLUKA with perpendicular incidence on the sensor. The vertical lines correspond to the detection threshold at 8 keV, below which pixel hits are missed, and the 2 MeV/cluster level, above which one sees evidence of saturation effects. The neutron histogram is affected by the low detection efficiency, and it is scaled up by a factor of 10 for visualization purposes.

The hadrons can also interact inelastically, in which case higher energy deposits are expected, often well beyond the saturation threshold, as clearly visible in Figure 16. Indeed, the simulations show that the largest energy depositions in the Timepix3 Radiation Monitor at the LHC are caused by inelastic nuclear reactions, which can be induced by neutrons and protons but also by pions. These could potentially be used to indirectly measure the HEH fluence (similarly to the RadMon detection mechanism [4]), either by operating with a high threshold, thereby filtering out the lower energy deposition events (i.e., the elastic interactions) or if the cluster properties would exhibit meaningful differences.

Finally, Table 2 quantifies the fraction of particle hits from different particles in the LHC mixed field, leading to energy depositions below the detection threshold of 8 keV in the linear calibration region, and above the 2 MeV cluster-level saturation threshold. In addition, the table also quantifies the fraction of the total ionizing dose (equivalent to the fraction of deposited energy) falling into the same three categories. For charged particles, the vast majority of the hits fall in the linear calibration regime, and most of the TID also fall into the same category (with the partial exception of protons, for which 24% of the TID is deposited by the small fraction of hits depositing more than 2 MeV each). Concerning photons, 22% of the hits fall below the detection threshold, but they only correspond to 2% of the TID and no hits are expected to be above the saturation limit. Lastly, neutrons yield a very large fraction of hits, leading to negligible amounts of TID, but while only a small fraction of the hits is above the saturation threshold, the latter (that correspond to inelastic collisions) are depositing a dominant fraction of the total TID. The results in Table 2 imply that the Timepix3 Radiation Monitor can be regarded as a reliable radiation monitor for the LHC mixed field, accurately measuring the count rate (hence, the flux) of charged particles and also measuring the associated TID with a linear response (with some saturation possibly occurring for protons). For the neutral particles, the measurement of photon-induced TID is also expected to be fully in the linear calibration regime, whereas saturation effects are expected to cause biases in the measure of the neutron-induced TID.

**Table 2.** The expected fraction of particle counts and fraction of the total ionizing dose (TID) deposited in the Timepix3 Radiation Monitor, as simulated in FLUKA, below the detection threshold (8 keV), in the linear calibration regime, and above the saturation threshold of 2 MeV.

| Particle | Below Threshold | | Linear Regime | | Saturation Effects | |
|---|---|---|---|---|---|---|
| | Counts [%] | TID [%] | Counts [%] | TID [%] | Counts [%] | TID [%] |
| Protons | $12 \times 10^{-6}$ | $1.14 \times 10^{-8}$ | 97 | 76 | 3 | 24 |
| Muons | $0.7 \times 10^{-3}$ | $7.51 \times 10^{-4}$ | 99 | 93 | 0.8 | 7 |
| Pions | $15 \times 10^{-3}$ | $8.16 \times 10^{-4}$ | 99 | 95 | 0.3 | 5 |
| Electrons | $12 \times 10^{-3}$ | $9.73 \times 10^{-4}$ | 99 | 99 | $1.25 \times 10^{-4}$ | $2.93 \times 10^{-3}$ |
| Photons | 22 | 2 | 78 | 98 | 0 | 0 |
| Neutrons | 92 | 1 | 7.4 | 28 | 0.4 | 71 |

## 7. Conclusions

Within the framework of R2E activities at CERN, a 300 μm thick silicon Timepix3 detector was characterized as a radiation monitor for the measurement of the mixed field of the LHC accelerator. The LHC radiation field includes different kinds of particles with broad energy spectra, all originating from the interaction of TeV-scale beam particles or secondary collision products, as simulated with the FLUKA Monte Carlo code.

A detailed description of the Timepix3 Radiation Monitor was given, covering the hardware setup and of the Timepix detection principle based on the simultaneous measurement of the time-over-threshold (ToT) and time-of-arrival (ToA). The charge released in the interaction of ionizing particles within the pixel matrix is typically spread over multiple pixels, leading to the need for a cluster reconstruction algorithm to identify the individual particle hits, for which the DPE [48] package has been used. From the literature, it is known that the Timepix calibration procedure, which translates the measured ToT per pixel into deposited energy, presents several regions of distinct dependency, including complex

saturation effects at high energy. To mitigate these effects, the Timepix3 Radiation Monitor is operated in hole collection mode at a partial bias voltage of 50 V, enhancing the charge sharing between pixels.

The energy calibration of the Timepix3 Radiation Monitor was carried out at the Centro Nacional de Acceleradores (CNA) using quasi-mono energetic beams of protons (alphas) from 0.6 (1) to 5 (5.6) MeV stopping in the active volume of the sensor. The energy deposition in the passive front layers (a known 500 nm metalisation layer and a silicon dead layer of an unknown thickness but experimentally determined) is computed using dedicated Monte Carlo simulations performed with FLUKA. The clusters formed by the CNA beams are reconstructed and filtered using dedicated shape parameters, ensuring that single particle hits are fully reconstructed and isolated. The cluster-level calibration analysis indicates that the detector operates in a linear regime with $ToT_{reg}(E_{dep}) = E_{dep} \cdot 993 \pm 93$ [25 ns/MeV] $+ N \cdot 23.10 \cdot 2.14$ [25 ns/pixel], while evidence of saturation effects is visible for energy depositions above 2 MeV, especially for the case of proton beams. At the pixel level, the saturation is estimated to occur from around 600 keV. The larger levels of saturation observed with protons compared to alphas are found to be associated with a different degree of charge sharing in the respective clusters, thus confirming that operating the sensor with a reduced bias voltage to enhance charge sharing is a good strategy to mitigate the saturation. No evidence of high-energy non-linear saturation or volcano effect is observed in the data. As part of the calibration measurements, the detector was irradiated at several angles up to 45° with 597 keV protons to estimate the dead layer thickness of the chip, quantified to be $333 \pm 60$ nm.

The Timepix3 Radiation Monitor can have several usages, similar to the existing monitoring technologies, but also complement them in the lower energy detection threshold, thereby possibly having more discriminating power. It can be used as a particle flux monitor for charged particles without discriminating their kinetic energy if they pass through the detector or fully reconstructing their energy if they stop within the active volume. Neutral particles have a lower interaction probability, for which conversion layers (e.g., LiF [58]) could be envisaged. By studying with FLUKA the energy deposited in the sensors by particles in the LHC mixed radiation field, the detector is found to be suitable as a dose rate monitor since the particles are expected to deposit energy within the linear calibration regime, with only a small fraction of the hits leading to saturation effects. As a partial exception, saturation is expected to play a non-negligible role when measuring neutron-induced TID due to the large energy deposited by inelastic nuclear reactions. Nevertheless, the observed performance of the Timepix3 Radiation Monitor opens the door for its usage for radiation field measurements at the CERN accelerator complex.

**Author Contributions:** Data acquisition, D.P., I.S. and D.L.; experimental facility coordination, P.M.-H., A.R.-M. and Y.M.G.; formal analysis, D.P.; Timepix setup, H.S. and J.S.; writing—original draft preparation, D.P.; writing—review and editing, D.P., G.L. and R.G.A. All authors have read and agreed to the published version of the manuscript.

**Funding:** This work has been sponsored by the Wolfgang Gentner Programme of the German Federal Ministry of Education and Research (grant No. 13E18CHA).

**Institutional Review Board Statement:** Not applicable.

**Informed Consent Statement:** Not applicable.

**Data Availability Statement:** Publicly available data sets were analyzed in this study. These data can be found here: https://cernbox.docs.cern.ch/ (accessed on 2 January 2024).

**Acknowledgments:** We thank the coordinators and operators of the CNA facility during our test campaign. Special thanks are given to Natalia Emriskova, for useful discussions about pixel detectors, to Lukas Tlutos from the Medipix collaboration and to Hubert Kroha, the university supervisor of Daniel Prelipcean.

**Conflicts of Interest:** The authors declare no conflicts of interest.

**Abbreviations**

The following abbreviations are used in this manuscript:

| | |
|---|---|
| ADC | Analogue-to-Digital Converter |
| ASIC | Application-specific integrated circuit |
| BLM | Beam Loss Monitor |
| CERN | European Organization of Nuclear Research |
| CNA | Centro Nacional de Acceleradores |
| DOFRS | Distributed Optical Fibre System |
| FPGA | Field Programmable Gate Array |
| fToA | fast Time-of-Arrival |
| LET | Linear Energy Transfer |
| LHC | Large Hadron Collider |
| LVDS | Low Voltage Differential Signaling |
| MIP | Minimum Ionizing Particle |
| PCB | Printed Circuit Board |
| R2E | Radiation to Electronics |
| SEE | Single Event Effect |
| SFP | Small Form-factor Pluggable |
| ToA | Time-of-Arrival |
| ToT | Time-over-Threshold |

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
