# Peer review of "Towards a Timepix3 Radiation Monitor for the Accelerator Mixed Radiation Field: Characterisation with Protons and Alphas from 0.6 MeV to 5.6 MeV"

_applsci, doi:10.3390/app14020624_

Round 1

Reviewer 1 Report

Comments and Suggestions for Authors

The paper presents the results of the Timepix multipixel Silicon particle detector characterisation using MeV H and He ion beams. Ions were used to inject charge in the device at different depths and angles, with different energies. Energy calibration and dead layer estimate were derived.

The research motivation and goals have been presented clearly - the need to investigate the parameters of operation of this detector for radiation monitoring in CERN, however, the methodology and presentation of the results need to be improved. Specifically these points:

1. Sections 3.2 and 3.3 describe the signal formation in the detector, and state that the hole drift is the main contributor to the induced signal, as well as explain the charge sharing between pixels due to diffusion. The paper must provide a schematic structure of the active volume of the detector and relate the previous description to the electric field and charge collection process in lateral and longitudinal dimensions. The reader is referred to Ref. 36 which is not available. 

2. The methodology of the dead layer calculation is described in lines 530 - 547, and is somewhat precarious. Was it possible to approach the problem of dead layer estimate from angular charge injection in a more simple procedure: If one calculates ΔE/E_0 = (E(θ) - E(θ=0))/E(θ=0) and utilises ion transport code to estimate energy loss in aluminium dead layer (known) and dE/dx at the interface of Al and Si (assuming that the stopping power is nearly constant in the dead layer) - plot of ΔE/E_0 vs. (1-1/cosθ) will allow for direct estimation of the dead layer thickness. Please comment if this approach is possible, and, if yes, compare it to yours.

Minor comment: Figure 10, bottom left plot, 5.6 MeV fit does not represent well the data.

Reviewer 2 Report

Comments and Suggestions for Authors

The paper describes a test campaign with quasi-mono energetic protons (alphas) from 0.6(1) to 5  (5.6) MeV and a calibration of the Timepix3 detector in this energy region.

 The main results:

As a result, the detector is found to be suitable for measuring particles in the LHC mixed radiation field within the linear calibration regime, with the partial exception of inelastic nuclear reaction hits (mostly from neutrons).

Comments.

The spectra presented in the Fig. 1 show a bright energy region of particles produced due to various collisions:

Neutrons: from 10-13  to 10 GeV

Mesons and pions: from 10-4 to 100 GeV

Gamas: from 10-4 to 1 GeV

The analyses were done for energies from ~1 to 6 MeV. Therefore, for energies above 6 MeV this detector, very likely, will produce just a constant signal, because the particles (protons and alphas) will not stop in the detector.

Therefore, the results for energies above 6 MeV cannot be justified as correct. This detector can be actually used just in lumped mode, that is, to detect whether we have some particles. It cannot be used to restore the energy spectrum. The responses produced for different particles with different energies will be overlapped producing at the end a spectrum of signals that cannot be resolved. As for neutrons and gammas this detector, as stated in the paper, is not quite useful.  In case of high energies (protons and alphas) a sequence of such detectors (one behind another) should be used for a accurate detection.

I suggest to make clear statements on the practical use of this detector: which particles, which energies, purpose of its use.

Round 2

Reviewer 1 Report

Comments and Suggestions for Authors

The authors have provided the answers to my questions:

-For the first question, the added schematic does help to understand the charge transport in the detector better. The updated text of the manuscript also provided some more information. A more substantial description would be beneficial, but this is sufficient.

- For the second question, the authors provided a detailed response. The suggested approach to dead layer estimate would result in a linear fit to the data, with the slope of the fit being the dead layer, but without the complications of the iterative process. The authors did not recognise and apply the idea completely, but I believe that the calculation and comments give enough proof that the dead layer estimate is accurate enough.

I believe the authors considered and addressed my questions, and the paper can be published.

Reviewer 2 Report

Comments and Suggestions for Authors

It fine for me.